# The developmental relationship between teeth and dermal odontodes in the most primitive bony fish *Lophosteus*

Donglei Chen[1]*, Henning Blom[1], Sophie Sanchez[1,2,3], Paul Tafforeau[3], Tiiu Märss[4], Per E Ahlberg[1]*

[1]Department of Organismal Biology, Uppsala University, Uppsala, Sweden; [2]SciLifeLab, Uppsala University, Uppsala, Sweden; [3]European Synchrotron Radiation Facility, Grenoble, France; [4]Estonian Marine Institute, University of Tartu, Tallinn, Estonia

**Abstract** The ontogenetic trajectory of a marginal jawbone of *Lophosteus superbus* (Late Silurian, 422 Million years old), the phylogenetically most basal stem osteichthyan, visualized by synchrotron microtomography, reveals a developmental relationship between teeth and dermal odontodes that is not evident from the adult morphology. The earliest odontodes are two longitudinal founder ridges formed at the ossification center. Subsequent odontodes that are added lingually to the ridges turn into conical teeth and undergo cyclic replacement, while those added labially achieve a stellate appearance. Stellate odontodes deposited directly on the bony plate are aligned with the alternate files of teeth, whereas new tooth positions are inserted into the files of sequential addition when a gap appears. Successive teeth and overgrowing odontodes show hybrid morphologies around the oral-dermal boundary, suggesting signal cross-communication. We propose that teeth and dermal odontodes are modifications of a single system, regulated and differentiated by the oral and dermal epithelia.

**\*For correspondence:**
donglei.chen@ebc.uu.se (DC);
per.ahlberg@ebc.uu.se (PEA)

**Competing interests:** The authors declare that no competing interests exist.

## Introduction

A tooth is a particular type of 'odontode' (*Figure 1A*): an exoskeletal structure that forms at an interface between an epithelial fold and the underlying mesenchyme, by dentine growing inwards from the epithelial contact surface. A mature odontode consists of dentine, in some cases covered with enamel or other hypermineralized surface tissue, irrigated through a central pulp cavity or pulp canals, and attached by a vascularized bone-like tissue (*Ørvig, 1977*; *Smith and Hall, 1993*; *Huysseune and Sire, 1998*; *Karatajute-Talimaa, 1998*). Although teeth are the only odontodes to persist in tetrapods (*Reif, 1982*), various forms of dermal odontodes covered the entire body surface of many extinct jawless vertebrates, before jaws and teeth evolved (*Janvier, 1996*), and persist in some extant groups. Probably, the most familiar examples of dermal odontodes in extant vertebrates are the placoid scales of sharks, which are commonly called 'skin teeth'. Besides this form of individual pointed denticles, multiple dermal odontodes can be fused to a basal plate, like in the growing scales of primitive chondrichthyans (*Reif, 1978b*), or anchored on a dermal bone, like on the skull bones of primitive osteichthyans, where the odontodes are usually accreted into patterned tubercles or ridges, referred to as 'ornament'. Dermal odontodes have long been regarded as an independent developmental module distinct from teeth, because of their supposed lack of the temporo-spatial regulation that, in teeth, is provided by a specific epithelial structure such as a dental lamina or odontogenic band (*Reif, 1982*; *Fraser and Smith, 2011*). However, shark placoid scales were recently shown to be patterned by a Turing-like mechanism (*Maisey and Denton, 2016*;

**eLife digest** Human teeth are an example of odontodes: hard structures made of a material called dentine that are sometimes coated in enamel. Teeth are the only odontodes humans have, but other vertebrates (animals with backbones) have tooth-like scales on their skin. These structures are called dermal odontodes, and sharks and rays, for example, are covered with them.

How these structures evolved, and whether teeth or dermal odontodes developed first, continues to spark great discussion among palaeontologists. Some researchers think that teeth evolved from dermal odontodes, a theory known as the 'scales-to-teeth' hypothesis. Others think dermal odontodes are distinct from teeth because they lack the same spatial organization. To investigate this further, palaeontologists are looking at the earliest examples of odontodes they can find: fossils of early vertebrates that carry both teeth and dermal odontodes.

Here, Chen et al. have studied *Lophosteus*, one of the earliest bony fishes that lived more than 400 million years ago, to explore early tooth evolution and growth patterns. Chen et al. digitally dissected a fossilized *Lophosteus* jawbone using submicron X-ray imaging, a technique with resolution to less than one millionth of a metre. Imaging thin sections of the specimen, found in Estonia, Chen et al. reconstructed the entire sequence of odontode development in the bony fish in 3D.

The analysis showed that teeth and dermal odontodes initially take shape together but differentiate as they grow, presumably instructed to do so by various developmental signals. However, at a later stage, the two types of odontodes become similar in appearance again, suggesting that they respond to each other's signals. For example, as the jawbone grows, dermal odontodes overgrow the earliest formed teeth. These younger odontodes resemble teeth, while the new teeth developing near the dermal odontodes take after dermal odontodes.

These findings suggest that teeth and dermal odontodes are not wholly separate systems but, instead, are closely related on a molecular level. The results also show that contrary to the 'scale-to-teeth' hypothesis, teeth do not evolve from fully formed dermal odontodes, rather the two types of odontodes form out of one founder.

This research builds on our knowledge from modern sharks and points to a previously unrecognised evolutionary relationship between teeth and dermal odontodes. It also furthers our understanding of how molecular regulation controls development.

*Cooper et al., 2018*). Nevertheless, current knowledge about the growth patterns of dermal odontodes is basically limited to modern sharks.

The evolutionary developmental relationship between teeth and dermal odontodes is pivotal for understanding the origin of teeth. The classic 'outside-in' hypothesis is currently in favor after decades of debate (*Donoghue and Rücklin, 2014*; *Haridy et al., 2019*), but the developmental continuum between teeth and dermal odontodes, which is one of its central premises, still lacks unequivocal evidence. Extant gnathostomes with dermal odontodes (sharks, rays, and some bony fishes such as *Polypterus*) always display a sharp demarcation between teeth and ornament. Even though they can provide data of all ontogenetic stages, they are not informative about the evolution of the developmental relationship between teeth and dermal odontodes. For that we must turn to the fossil record of the earliest jawed vertebrates, in particular to the jawed stem gnathostomes and the stem osteichthyans, which form the common ancestral stock of Chondrichthyes + Osteichthyes and of Actinopterygii + Sarcopterygii, respectively (*Figure 1B*).

This study presents the marginal dentition of the Late Silurian (422 million years old; https://stratigraphy.org/timescale/) stem osteichthyan *Lophosteus superbus*, based on investigation by propagation phase-contrast synchrotron microtomography (PPC-SRμCT), which allows the dentition to be digitally dissected in 3D with sub-micrometer resolution and the dental ontogeny to be reconstructed (*Figure 2*).

The same technique has revealed the earliest osteichthyan-style tooth replacement, in the 424 million year old stem osteichthyan *Andreolepis* (*Chen et al., 2016*), and the most phylogenetically basal gnathostome dentitions (*Vaškaninová et al., 2020*). The latter occur in the Early Devonian armored fish known as 'acanthothoracids', including *Radotina*, *Kosorapis*, and *Tlamaspis* (*Figure 1B*).

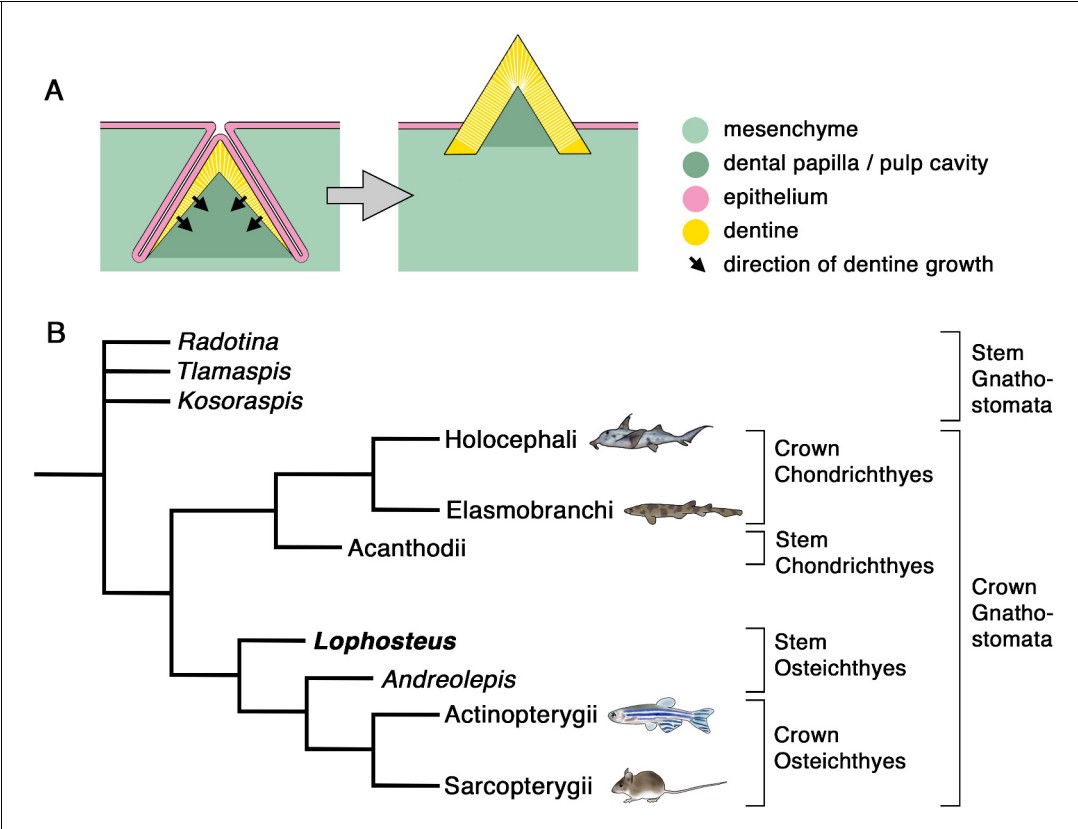

**Figure 1.** Odontode development and gnathostome phylogeny. (A) Schematic representation of developing (left) and mature (right) odontode. The odontode shown here lacks enamel, as do the teeth and dermal odontodes of *Lophosteus*. (B) Phylogenetic position of *Lophosteus* and some of the other fossil taxa discussed in this paper. The extant gnathostome groups are represented by developmental model organisms, as follows: Holocephali, *Callorhinchus milii*; Elasmobranchi, *Scyliorhinus torazame*; Actinopterygii, *Danio rerio*; Sarcopterygii, *Mus musculus*. Tree topology from *Qu et al., 2015* and *Vaškaninová et al., 2020*. Formal hierarchical categories indicated on right. Animal images from *Trinajstic et al., 2013*, except *Callorhinchus*, from *Ryll et al., 2014*, and *Scyliorhinus*, original.

They all have non-shedding dentitions with lingual tooth addition, carried by marginal dermal bones, suggesting that these are the ancestral conditions of teeth (*Vaškaninová et al., 2020*). Chondrichthyans and osteichthyans both inherited the lingual tooth addition, but evolved tooth shedding independently, while marginal jawbones ornamented by dermal odontodes were only kept by osteichthyans. The jawbones of *Kosoraspis* and *Tlamaspis* consist of multiple short pieces (*Vaškaninová et al., 2020*, Figs. 3 and 4), a condition strikingly similar to that in *Lophosteus* (*Figure 3A*, *Figure 3—figure supplement 1*), but unknown in other taxa. Stellate (star-shaped) dermal odontodes, which are characteristic of acanthothoracids, are also present in *Lophosteus* (*Figure 3—figure supplement 1*; *Figure 4A*, e.g. O3g-3-6) but not in *Andreolepis* or other described early osteichthyans. *Lophosteus* further resembles a stem gnathostome in completely lacking enamel, whereas *Andreolepis* has enamel on its scales (*Qu et al., 2015*). In fact, the only dental character of *Lophosteus* that unambiguously distinguishes it from acanthothoracids and identifies it as an osteichthyan is the presence of tooth shedding by partial and basal resorption (*Chen et al., 2017*). This character distribution suggests that *Lophosteus* is the least crownward of known stem osteichthyans (*Figure 1B*), making it uniquely informative about the evolution of the osteichthyan dentition.

## Results

A marginal jawbone of *Lophosteus*, GIT 760–12 (*Figure 3A*) from the Late Silurian (Pridoli) of Ohessaare Cliff, Saaremaa, Estonia, was scanned with an isotropic voxel (3D pixel) size of 0.696 μm. This

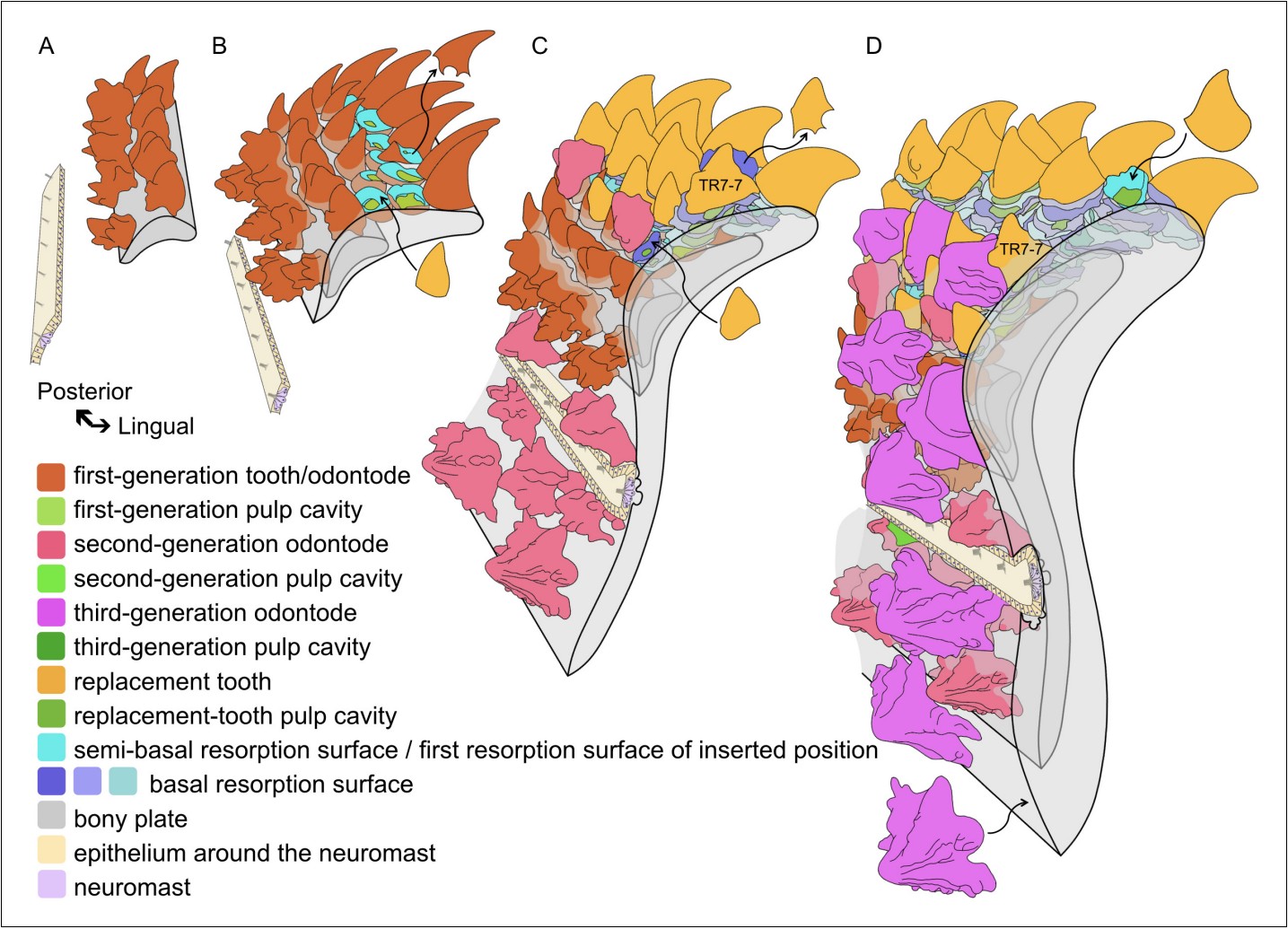

**Figure 2.** Reconstructed ontogeny of the *Lophosteus* marginal jawbone relative to the development of the lateral line. Block diagrams in antero-external view, aligned so that the biting margin maintains a fixed position (note that this causes the older parts of the bone to move away from the biting margin and rotate anticlockwise during growth). The founder ridges and the ornament-like tooth TR7-7 mark the labial rotation and drift of the tooth rows. (**A**) The initial odontodes are formed as founder ridges. (**B**) Isolated dermal odontodes and teeth are added sequentially in opposite direction, respectively attaining a stellate and conical morphology. The teeth are shed semi-basally, establishing replacement tooth positions. (**C**) Teeth are cyclically replaced at the positions set up by the first-generation teeth; the second-generation odontodes invade the oral domain lingually and form around the lateral line canal labially. Lines of resorption around the lateral line canal indicates bone remodeling (see *Figure 4—figure supplement 6*). Note, the replacement of teeth and the overgrowth of odontodes may have commenced before the addition of the ultimate first-generation teeth. (**D**) The ventral extension of the lateral line canal partially resorbs the second-generation odontodes at its ventral border; more tooth rows are overgrown by the third-generation odontodes after being rotated to the face; new tooth positions are inserted to compensate the embedded labial tooth positions. Curved arrows, examples of the adding or shedding of teeth or dermal odontodes. The buried part of teeth and dermal odontodes, the embedded resorption surfaces and bone mineralized at earlier stages become increasingly grey. The lateral line is represented by neuromasts and epithelial cells. The size and number of neuromasts is schematic and only represents their presence. Perspective view. Scale bar not applicable.

specimen is probably from the lower jaw as it carries a lateral line groove that parallels the jaw margin. It can be divided into oral and facial laminae, which form a transverse angle of about 130° (*Figure 3C*; *Figure 3—figure supplement 2*). The ossification center is located at the boundary of the two laminae, the thickest point of the bony plate. The feeder vessels of the odontode layer, which penetrate the bony plate with an increasing diameter and obliquity from lingual to labial, all radiate from here (*Figure 3C*, sky blue; *Figure 4—figure supplement 1B* and *Figure 5A*, arrows). A layer of large cuboidal osteocyte lacunae (*Figure 3—figure supplement 2B*; *Figure 4—figure supplement 2*) may represent the oldest part of the jawbone (*Figure 3—figure supplement 2A*, dashed curve) that was deposited rapidly during the earliest developmental stage (*Davesne et al., 2020*).

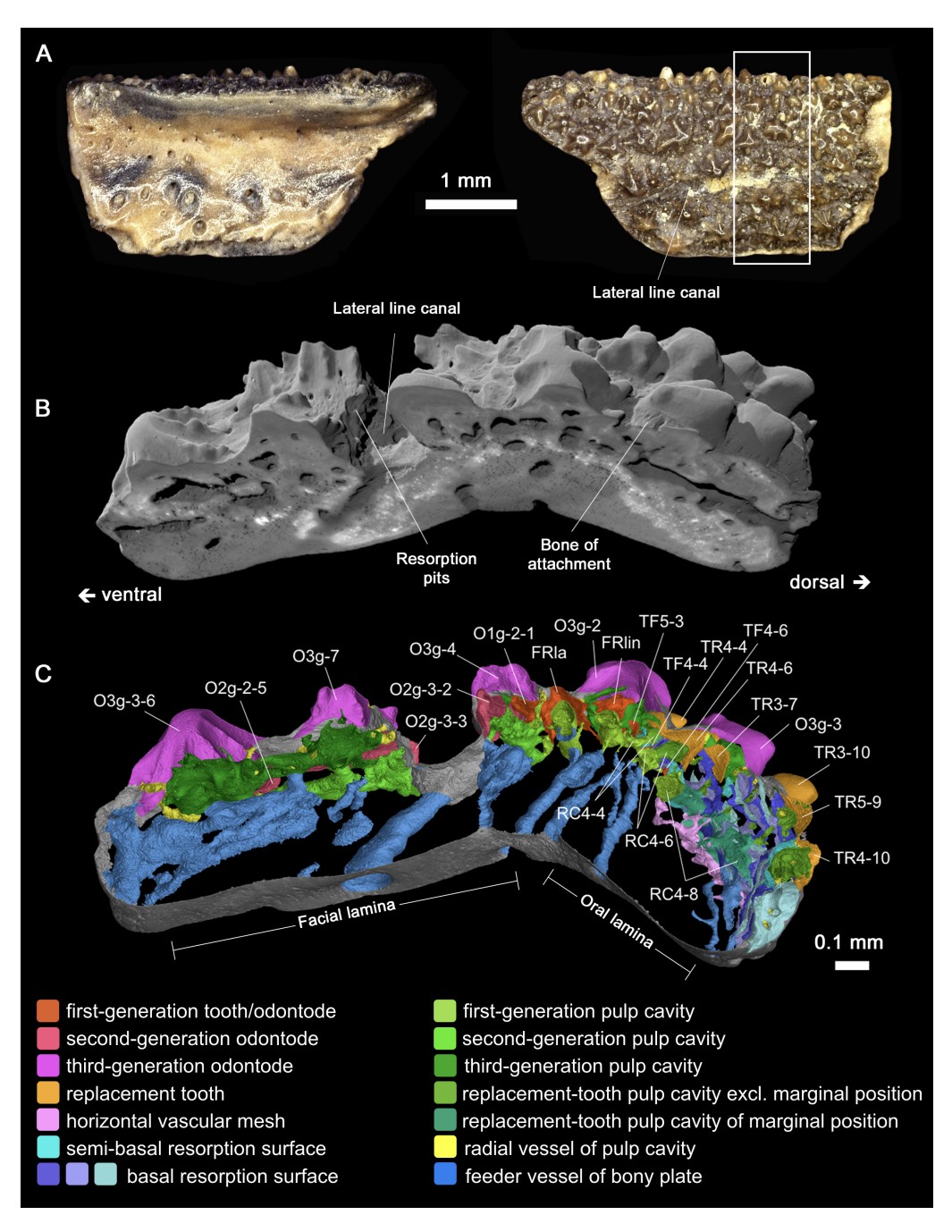

**Figure 3.** Gross morphology and digital disection of the *Lophosteus* marginal jawbone GIT 760–12. (**A**) Photographs oriented with biting (presumed dorsal) margin at the top, in internal (left) and external (right) views. White box indicates region where internal structures have been fully modeled. (**B**) Antero-external view of a perspective block. (**C**) Antero-visceral view of a slab of the 3D histologic model through the midline of File 4. The most labial tooth positions have the shortest replacement history (TF5-3 has not been replaced); the marginal tooth positions have the longest replacement history (RC4-8 has been replaced 17 times), while the inserted tooth positions immediately labial to the marginal ones have the most recent history (TR4-10 has been replaced once).

The online version of this article includes the following figure supplement(s) for figure 3:

**Figure supplement 1.** Morphologic variation of the marginal jawbones in *Lophosteus*.

**Figure supplement 2.** Virtual transverse sections between File 3 and 5.

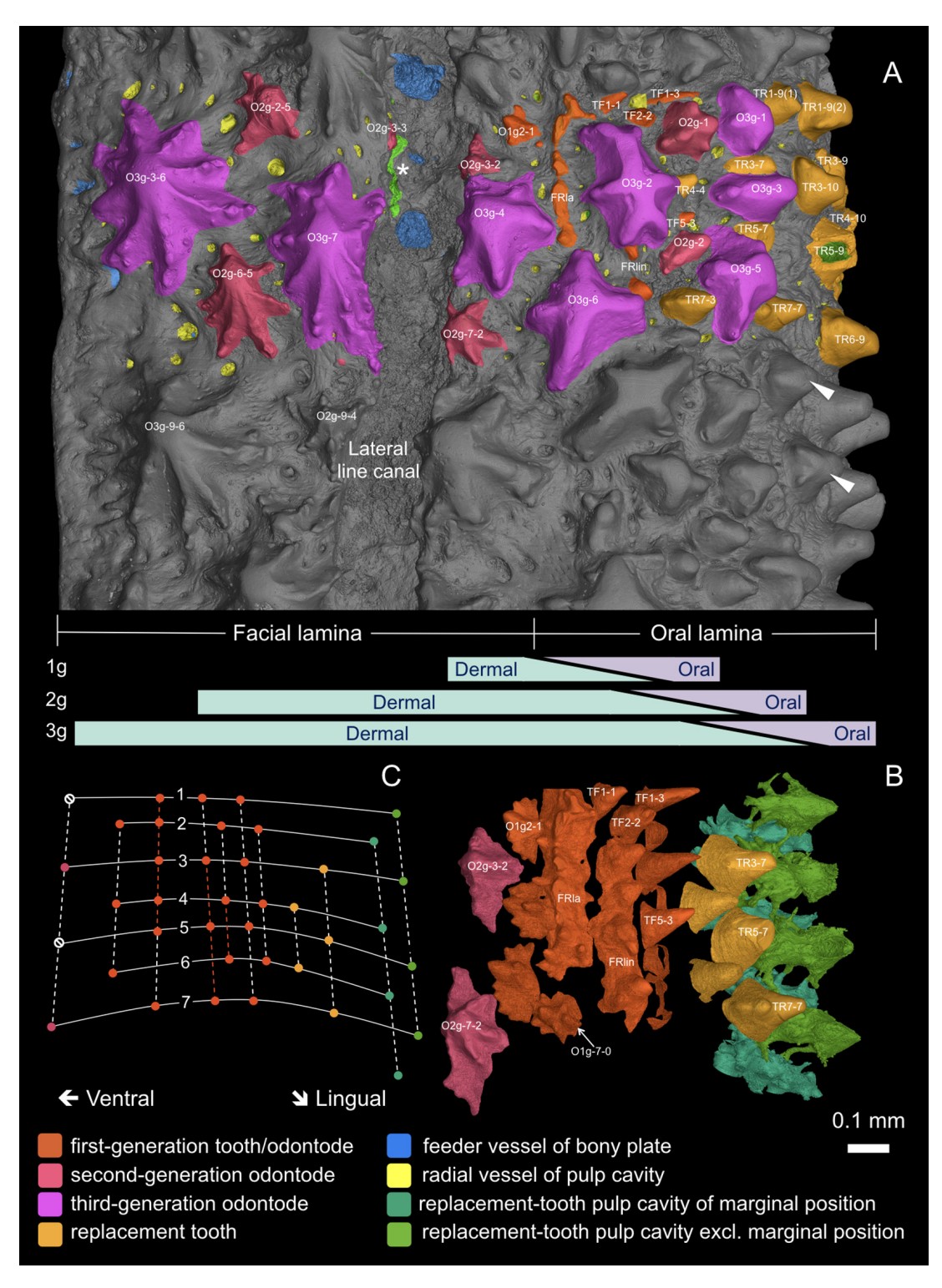

**Figure 4.** 3D external view of the scanned area. For three-dimensional curvature, see *Figure 3*. (A) External morphology with individually modeled structures highlighted in color. White arrowheads point to ornament-like teeth with side-cusps in the uncolored area. Bars in lavender and mint green indicate the putative gradient of the oral and dermal signal spheres, and the shift of the oral-dermal boundary during deposition of the first-generation (1g), second-generation (2g) and third-generation (3g) odontodes. (B) Overgrowing odontodes and bone matrix rendered invisible to show a consistent alternate pattern between replacement teeth, first-generation teeth and dermal odontodes dorsal to the lateral line canal. The replacement teeth at the inserted positions are not shown. Because the lingual rows of first-generation teeth are obscured by the labial rows of replacement teeth in this view,

*Figure 4 continued*

these rows are not shown (for all tooth rows, see *Figure 5B*). For optimal visibility, the most lingual rows of replacement teeth are represented by their pulp cavities. (**C**) Diagram of the alternate organization based on B. Solid lines, transverse files; numbers of files mark the putative level of the ossification center. Dashed lines, longitudinal rows; colored parts of the dashed lines indicate the range of the founder ridges. Dots denote positions of the structures in the particular colors. Note, the second-generation odontodes situated along the lingual border of the lateral line canal (O2g-3–2, O2g-7–2) have their labial ridgelets truncated, in order not to intrude on the canal sulcus. The alternate positions that are supposed to form the next row are suppressed (null signs), but the same alternate pattern develops properly on the other side of the canal (see *Figure 4—figure supplement 1B*). The dorsal edge of the odontodes at the ventral border of the canal is resorbed, leaving their pulp cavities open to the canal (asterisk; see *Figure 3*, O2g-3–3; *Figure 4—figure supplement 1B*, OP3-3, OP7-3; *Figure 4—figure supplement 4*). The ridgelets of the third-generation odontodes at both borders of the canal are also compressed (O3g-4, O3g-7).

The online version of this article includes the following figure supplement(s) for figure 4:

**Figure supplement 1.** 3D external views (same as *Figure 4*) of the vascular system and the organization of facial ornament.
**Figure supplement 2.** Slideshow of serial virtual thin sections — Slice A.
**Figure supplement 3.** Slideshow of serial virtual thin sections — Slice B.
**Figure supplement 4.** Slideshow of serial virtual thin sections — Slice C.
**Figure supplement 5.** Slideshow of serial virtual thin sections — Slice D.
**Figure supplement 6.** Slideshow of serial virtual thin sections — Slice E.

The odontode component of the jawbone consists of an extremely complex multi-layered assemblage of dermal odontodes, teeth (complete or partly resorbed), pulp cavities, and resorption surfaces (*Figure 3C*). Although the complexity of the data set necessitates a very detailed description to arrive at an understanding of the patterning and growth processes, the actual process of interpretation is conceptually straightforward. It rests in essence on four principles. Firstly, that if one odontode overgrows another, the overgrowing odontode is younger than the overgrown; secondly, that a resorption surface is younger than the structures it cuts into; thirdly, that a series of stacked resorption surfaces centered on a single pulp cavity indicates persistence of a single tooth position with repeated replacements of the tooth; and fourthly, that odontode files can be identified by the geometric alignment of odontodes on the jawbone. These criteria allow us to reconstruct the complete sequence of ontogenetic events (*Figure 2*; *Video 1*).

Here, we present a brief overview of the main ontogenetic stages, before proceeding to the comparative analysis in the Discussion; additional descriptive details are given in the figure legends. Odontodes that are directly attached to the oldest part of the jawbone are designated as 'first-generation', even if they partly overlap the positions of previously resorbed first-generation odontodes. The first generation thus does not represent a single moment in time, but rather the initial process of establishing the odontode array of the jawbone. Each standing odontode, including the ornament (O), first-generation teeth (TF) and replacement teeth (TR), is numbered according to its location and generation. Pulp cavities are numbered to denote ornament positions (OP), tooth positions (TP) and replacement columns (RC), marked as 'file number-row number'. A replacement column is formed by the succession of teeth at a tooth position. Replacement columns that derive directly from a first-generation pulp cavity have the same number as the first-generation teeth. The youngest inserted positions are not aligned in rows, but all of their row numbers are set as 10 in order to be differentiated from the others. The generation membership of standing replacement teeth varies dramatically between tooth positions, and thus is not indicated (though it can be calculated from the number of resorption surfaces); but dermal odontodes are distinguished by generation numbers (1g/2g/3g), which is added before the file numbers. For example, O2g-6–5 represents the second-generation odontode that is situated in the sixth file of the fifth odontode row. File numbers can be overviewed from *Figure 4C* (7 files in the modeled area), and generation number are illustrated by colors. The file/row numbers are not applicable to the overgrowing odontodes that are located irregularly, and 'generation number-serial number' is used instead. For instance, the '7' of O3g-7 is neither related to File 7 nor Row 7.

## Primordial odontodes

The oldest odontodes in the entire system are two rows of first-generation odontodes, fused into longitudinal ridges with confluent pulp cavities, at the level of the probable ossification center (*Figures 2A*, *3C* and *4B*; *Figure 3—figure supplement 2*, FRla, FRlin). We designate them as

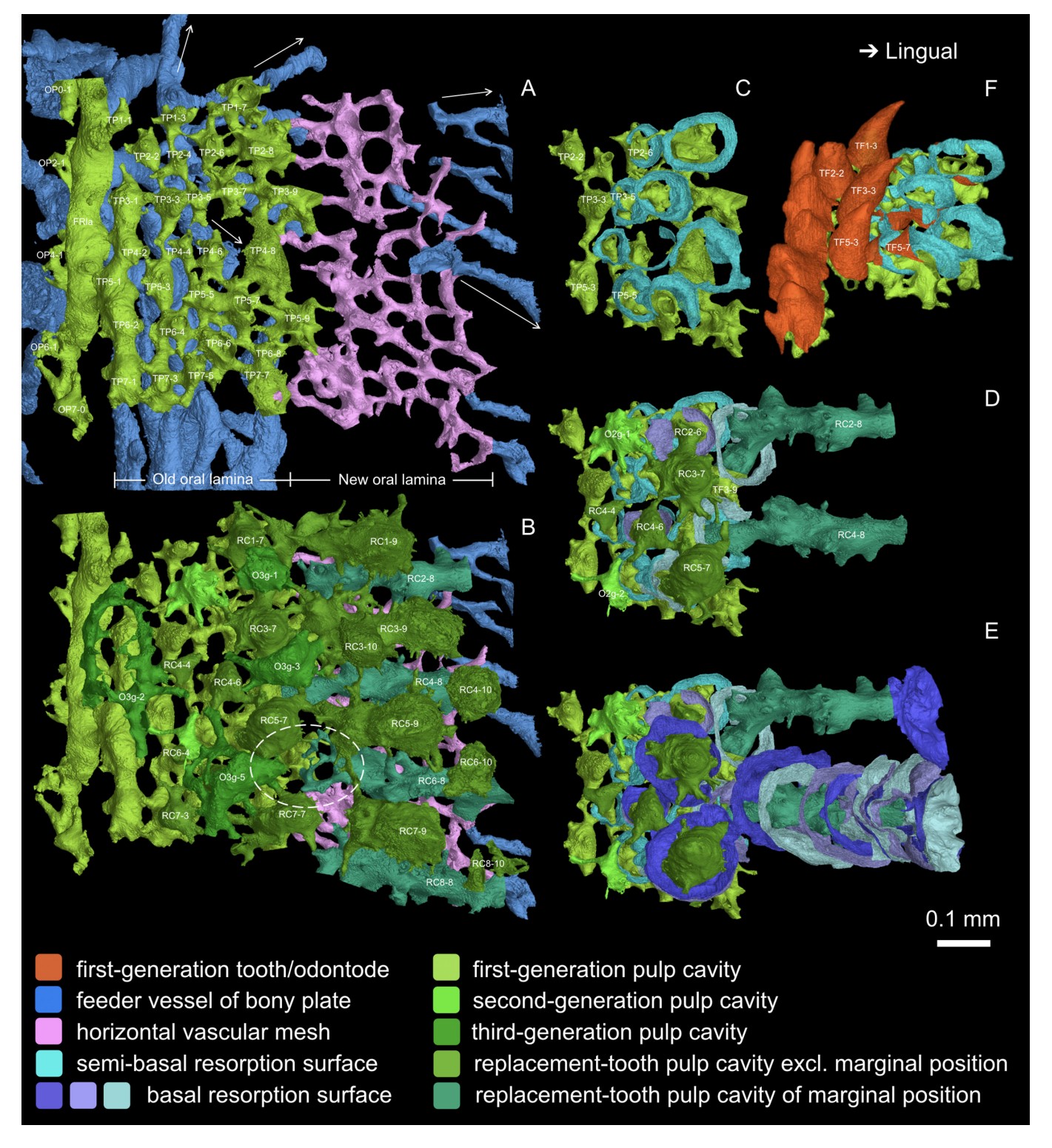

**Figure 5.** Occlusal view of the oral lamina. (A-B) Numbering of tooth positions (TP). (A) Pulp cavities of the first-generation teeth and dermal odontodes. Arrows indicate the directions of the feeder vessels radiating from the ossification center. The feeder vessels in blue run longitudinally only beneath the first-generation teeth, but never do it labially, and these longitudinal feeder vessels may have penetrated the old oral lamina at the early stage. The horizontal vascular mesh in pink, which represents the new oral lamina beyond the first-generation teeth, supports the lingual replacement columns in the interval that lacks feeder vessels. (B) Pulp cavities of replacement teeth and overgrowing odontodes are added. For the feeder vessels,

*Figure 5 continued on next page*

*Figure 5 continued*

only those newly incorporated at the jaw margin are shown. Note, RC2-6 is covered by O3g-1, and RC6-6 is covered by O3g-5. Dashed oval, an example of the discontinuity of the pulp cavities between a first-generation tooth and its successive replacement teeth, which suggests the replacement of the first-generation tooth has been delayed, but the drift of the delayed replacement teeth still follows the same file. All inserted positions are also aligned with the tooth files of preexisting positions. (C-E) Comparison between the tooth replacement of File 2–3 and File 4–5. C is aligned to A. The successive resorption surfaces of RC2-4, which are similar to those of RC4-8, are not shown, except the last and the first basal resorption surface and the semi-basal resorption surface. Note, the resorption surfaces of RC4-8 gradually change in orientation and density. (F) Antero-occlusal view of the lingual founder ridge and the first-generation teeth showed in C, showing the transition from no resorption, via semi-basal resorption, to basal resorption, as the tooth rows increasingly overlap.

'founder ridges'. On the lingual founder ridge, the main cusps are tall, conical, lingually recurved and widely spaced; on the labial founder ridge they are blade-like, labially inclined and united by small cusps. Side-cusps are more numerous on the labial founder ridge. The labial flanks of both ridges carry more side-cusps than the lingual flanks (*Figures 4B* and *5F*). There is no overlap between the two ridges; instead, their bases join as a continuous sheet (*Figure 3—figure supplement 2A*), implying they formed simultaneously.

Following the establishment of the founder ridges, more isolated odontodes were added sequentially in both lingual and labial directions, overlapping the lingual and labial edges of the founder ridges, respectively (*Figure 3—figure supplement 2*, O1g-2–1 and TF5-3). These new odontodes are unicuspid, conical teeth on the lingual side of the lingual ridge (*Figure 5F*), but are multicuspid and quickly take on the stellate morphology with crenulated ridges on the labial side of the labial ridge (*Figure 4B*). Simply put, as the odontode skeleton spreads away from the two founder ridges, it turns into teeth lingually and into ornament labially.

## Teeth

In the rows that follow lingually from the lingual founder ridge, all the main cusps become isolated and sharp. Although variable in size, they are arranged in semi-regular alternate files, and the lingual founder ridge can be regarded as a union of two alternate rows (*Figure 4B,C*; *Figure 5A*, TP3-1, TP5-1, TP7-1 and TP4-2, TP6-2). These unicuspid odontodes are considered as the first-generation teeth. Only the most labial (and thus oldest) first-generation teeth are complete. More lingually, the first-generation teeth are resorbed semi-basally, and their remaining bases overlap considerably (*Figures 2B* and *5F*), just like the first-generation teeth of the tooth cushions that form the inner dental arcades (*Chen et al., 2017*). The resorption surfaces are wide open, not only to the next first-generation tooth to be added to the file, but also to the replacement tooth buds that formed immediately above them (*Figure 5C–F*). The first-generation teeth thus establish the tooth positions for the cyclic replacement teeth

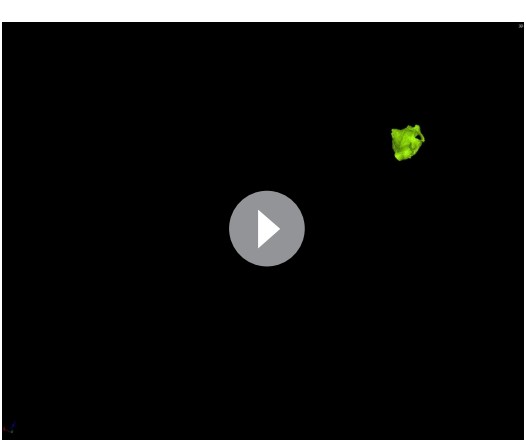

**Video 1.** Tooth addition and replacement of two alternate files. Perspective view. The order of the structures appearing in the video is not strictly consistent with that in life. However, the sequence of developmental events can be inferred: except for the most labial ones, each first-generation tooth had been resorbed before the next, more lingual, tooth was added to the file. As the successive tooth positions in a file increasingly overlap, at a point, semi-basal resorption becomes completely basal, and the successive teeth in a file now take on the appearance of a replacement column. The marginal position is drifting and tilting lingually, with the basal resorption surfaces become increasingly tighter. The addition of new first-generation teeth to the lingual ends of tooth files, and the deposition of replacement teeth onto the labial tooth sockets of these files, are likely parallel processes. The former process constructs tooth files from the positions set up by the founder ridges via semi-basal resorption, and the latter builds replacement columns at each position via basal resorption. The replacement cycles of the labial positions are terminated by the overgrowth of ornament. But where space allows, a new position can be inserted into the marginal replacement column, substituting the overgrown positions.
https://elifesciences.org/articles/60985#video1

(*Figure 2C,D*; *Video 1*). The pulp cavities of the first-generation teeth lie directly on the basal compact bone, coinciding with the territory of the basal feeder vessels (*Figure 5A*). This indicates that the most lingual row of first-generation teeth was once located at the jaw margin and that the oral lamina at this early developmental stage was much narrower (*Figure 2B*; *Figure 4—figure supplement 1*, TF).

## Ornament

Following labially from the labial founder ridge is a single row of isolated spiny odontodes (*Figures 2B*, *4B* and *5A*). They overlap the labial edge of the labial founder ridge substantially, without partially resorbing it. As a result, their pulp cavities are constrained by the space available (*Figure 3—figure supplement 2*, O1g-2–1; *Figure 4—figure supplement 1B* and *Figure 5A*, OP0-1, OP2-1, OP4-1, OP6-1). As the facial lamina extends labially, younger generations of odontodes are initiated to cover the new bone; as the jawbone rotates labially, they also begin to invade the oral lamina (*Figure 2C,D*; *Figure 4—figure supplement 1*, O2g, O3g; *Video 1*). Unlike the teeth, no resorption occurs in the dermal odontodes. Younger generations of larger odontodes thus simply overgrow, rather than replace, the preexisting dermal odontodes. There are two levels of overgrowth and the dermal odontodes are divided into three generations, inferred from their size and distribution.

All generations of dermal odontodes follow a global morphologic gradation. The further they are from the biting margin at the time they are initiated, the more side-cusps, ridgelets, and crenulations are associated with the main cusp, and the more ascending canals are attached to the flattened pulp cavities. The largest and youngest stellate odontodes have the most elaborate appearance (*Figure 4A*, O3g-3–6, O3g-7) and the most osteodentine-like tissue (*Figure 4—figure supplements 2–5*), which probably reflects their location furthest from the jaw margin, rather than their size or age. We surmise that this pattern reflects a morphogenetic signal gradient (*Figure 4*) between the jaw margin and the 'generic dermal surface' of the face, represented here by the facial lamina beyond the lateral line canal.

## Interaction between teeth and ornament

Where the dermal odontodes invade the tooth-bearing oral lamina, a subtle but important morphologic variability affects the labial surface of some shedding teeth. The replacement teeth at the marginal positions, like the first-generation teeth, always have a smooth labial surface, but teeth close to the invading ornament often carry side-cusps on the labial face (*Figure 4A*, arrowhead; *Figure 6C, D*, TR1-9(2), TR3-7, TR3-10, TR6-9, TR7-3, TR7-7). The dentine of the side-cusps tends to include some cell spaces. In other words, these teeth show ornament-like characteristics; there appears to be a degree of 'morphologic cross-contamination' between the two odontode sets (*Figure 4A*, bars in lavender and mint). At the invasive front line, both overgrowing odontodes and replacement teeth have a half-tooth half-ornament morphology (*Figure 6C*, compare O2g-1 and O2g-2 with TR7-3 and TR7-7), though the replacement teeth are recognizable by their possession of basal resorption surfaces.

## Discussion

### Comparative context: the dentitions of *Andreolepis* and 'acanthothoracids'

The marginal dentitions of the stem osteichthyan *Andreolepis* (Late Silurian, Gotland, Sweden) and the 'acanthothoracid' stem gnathostome *Kosoraspis* (Early Devonian, Prague Basin, Czech Republic) are of particular interest as comparators for *Lophosteus* because they show a similar combination of transverse alternate tooth files with lingual tooth addition, carried on marginal bones that also bear dermal odontodes on their facial laminae (*Chen et al., 2016*; *Vaškaninová et al., 2020*). *Lophosteus* aligns with *Andreolepis* and differs from stem gnathostomes in showing resorptive tooth shedding, a unique osteichthyan characteristic (*Chen et al., 2016*). Indeed, this process is of fundamental importance in shaping the dentition of *Lophosteus*, with some tooth positions showing as many as twenty replacement cycles. This has important implications for understanding the ontogeny underlying the final adult morphology.

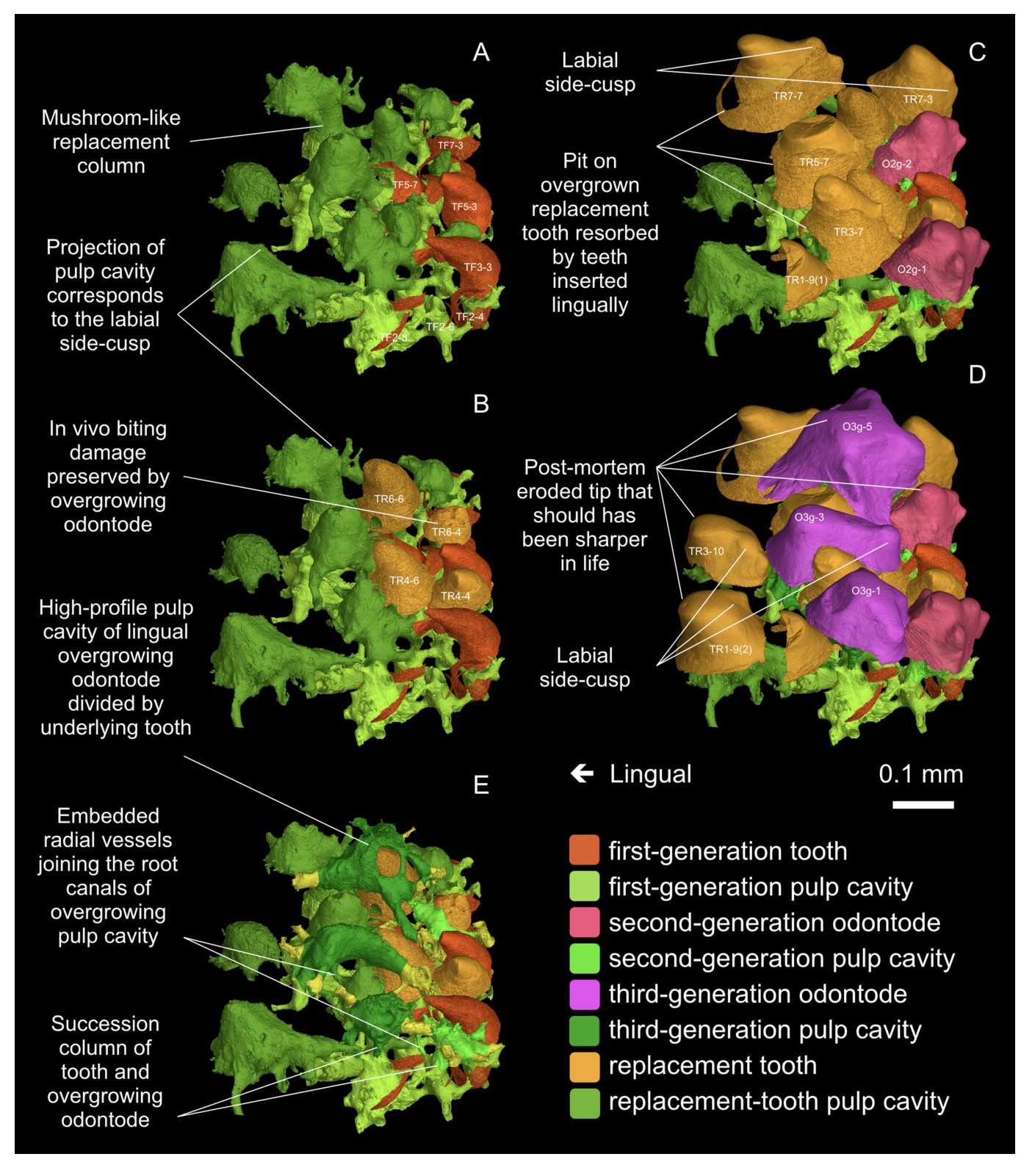

**Figure 6.** Postero-occlusal view of the tooth field invaded by overgrowing odontodes. (**A**) First-generation teeth are shown. They are perfectly conical. (**B**) The final replacement teeth that will be buried by second-generation overgrowing odontodes are shown. They are still perfectly conical. (**C**) The final replacement teeth that will be buried by third-generation overgrowing odontodes are shown. They are more or less ornament-like, probably because of the approach of the dermal epithelium, which is represented by the second-generation overgrowing odontodes, during the tooth development. (**D**)
*Figure 6 continued on next page*

*Figure 6 continued*

The final replacement teeth of the inserted positions are shown. They are ornament-like, probably because of the approach of the dermal epithelium that generate the third-generation overgrowing odontodes. The dermal odontodes overgrowing the oral lamina do not develop a fully stellate morphology, only with few uncrenulated ridgelets. The most lingual ones tend to become longitudinally compressed, contrary to the longitudinal elongation of those on the facial lamina. Their side-cusps only develop labially and main cusps incline lingually. Note, since the invasive front line of the ornament (the oral-dermal boundary) is undulating, tooth morphology is not correlated with rows or generations, but with the proximity of dermal odontodes. (E) Spatial relationship between the pulp cavities of overgrowing odontodes and the teeth buried right below. Note, the labial positions function at the earlier stages, depending on how soon they are overgrown by ornament, and thus the final replacement teeth of the labial tooth rows are smaller than those of the lingual ones. Each position, even if in the same row, has a different replacement history, which can be revealed by the shape of pulp cavity; in a long-life position, the first-generation tooth and all its replacement teeth have their pulp cavities fused into a mushroom-like shape.

*Kosoraspis* has a much simpler ontogenetic history without resorption-replacement cycles, essentially corresponding to the first-generation odontodes of *Andreolepis*. *Kosoraspis* shows a unidirectional change from small dermal odontodes at the external margin of the jawbone, through gradually larger and progressively more tooth-like dermal odontodes, to teeth (*Vaškaninová et al., 2020*). This indicates the ossification center is located at the external margin. The gradient between teeth and dermal odontodes of the first generation appears to be unidirectional in *Andreolepis* too, from a morphology with elongate bases to a more prominently tooth-like morphology with round bases as they approach the jaw margin (*Chen et al., 2016*, Figure 2a,b). Any equivalents of the founder ridges of *Lophosteus*, if present, have not been captured by the high-resolution scan that only covers approximately a quarter of the height of the facial lamina (*Chen et al., 2016*, Figure 1b). Nevertheless, it is certain that the location of the odontode founder region and the bone ossification center is considerably external to the original oral-dermal boundary in *Andreolepis*. By contrast, the three developmental landmarks overlap in *Lophosteus*, which thus displays a bidirectional morphologic gradation in the initial odontode skeleton.

## Developmental implications: initiation of the odontode skeleton

The dermal odontodes and teeth of *Lophosteus* are initiated simultaneously in the form of two parallel and closely spaced founder ridges (*Figures 2A,B* and *4B*). The cusps of the lingual founder ridge incline lingually, and those of the labial founder ridge, labially; subsequent odontodes are added to the lingual flank of the lingual founder ridge and the labial flank of the labial founder ridge. This geometric layout strongly suggests that the initiation site for odontode formation is the boundary between two patterning domains: a labial domain where the odontodes become ornament and a lingual domain where they become teeth. Because the development of an odontode is always initiated by an epithelium, we infer that the two domains are covered with two distinct epithelia, which we refer to respectively as the dermal and oral epithelium. The cusps of the two founder ridges are already morphologically distinct, and the odontodes that are subsequently produced at their labial and lingual sides quickly acquire the full characteristics of ornament and teeth, respectively. This must represent the establishment of their complete regulatory cascades, and possibly relates to the initiation of new odontodes on each side moving away from the boundary between the dermal and oral epithelia into the presumably more homogeneous signaling environment of a single epithelium (*Figure 4*, 1g).

## Developmental implications: the shifting oral-dermal boundary

The strict, linear separation described above only characterises the first-generation odontodes. Already in the second-generation, stellate odontodes have overgrown some of the oldest teeth, in a considerably more lingual position than the original boundary. The third generation penetrates even further into the territory of teeth (*Figure 4*, 2g and 3g; *Figure 4—figure supplement 1*, O2g, O3g). This suggests an irregular expansion of the dermal epithelium into the region previously occupied exclusively by the oral epithelium. A similar invasion of the tooth field by dermal ornament has been observed in *Andreolepis* (*Chen et al., 2016*) and an unnamed acanthothoracid from the Canadian Arctic, specimen CPW.9 (*Smith et al., 2017*; *Vaškaninová et al., 2020*).

In *Lophosteus*, the ornament invasion produces an effect that is highly informative about the relationship between these two odontode sets. Essentially, the *deposition* of odontodes remains

characteristic for the two sets – teeth continue to be replaced cyclically until overgrown by ornament, and the dermal odontodes are never shed – but the *morphology* of each set seems to become influenced by the other. The teeth nearest to the ornament bear side-cusps that tend to form a labial ridgelet. Conversely, in the most lingual dermal odontodes, the conical main cusp tends to stand out and point lingually, the side-cusps being restricted to the labial side (*Figures 4A,B* and *6C, D*). That is to say, in the invasion zone, the teeth are ornament-like and the dermal odontodes are tooth-like. The simplest explanation for this phenomenon is that the invasion zone provides a mixed set of morphogenetic signals, because it is a patchwork of dermal and oral epithelium, and that both dermal odontodes and teeth respond to both signals. Note, however, that this 'regulatory cross-contamination' only affects the morphology, not the deposition and (if present) resorption cycles.

In *Andreolepis*, the lingualmost odontodes of any generation of invading ornament can be tall and bear biting damage, in contrast to the characteristic flat-topped ornament morphology (*Chen et al., 2016*, Extended Data Figure 3c). In many basal actinopterygians with acrodin-capped teeth, such as *Birgeria* and *Boreosomus*, the dermal odontodes labial to the jaw teeth also bear an acrodin cap (*Ørvig, 1978a*, PP. 38, Figs. 3, 4, 7–9). The zone of the acrodin-bearing odontodes varies in size, being, for example, narrower in *Colobodus* and wider in *Nephrotus* and others (Ørvig, 1978c, pp. 307), but is invariably restricted to the vicinity of dentition, regardless of the type of jawbone (*Ørvig, 1978b*, pp. 41–42). Similar phenomena can be seen in stem chondrichthyans. The labial face of the blade-like teeth on the tooth whorls of *Ptomacanthus* can be ornamented, like the tesserae that they are interlocked with (*Miles, 1973*). The toothless *Obtusacanthus* displays a morphological gradation from stellate head scales, via fan-shaped scales, to tooth-like lip scales (*Blais et al., 2011*), comparable to the superficial morphological gradation from stellate facial ornament with crenulated ridgelets, via multicuspid invasive ornament and ornament-like replacement teeth, to unicuspid marginal teeth in *Lophosteus*. The labio-lingual rows of 'extra-oral teeth' on the whorl-like cheek scales, pointed lip scales, platelets or tesserae labial to the jaw margin of ischnacanthid acanthodians (*Gross, 1971*, Tafel 4, Fig. 24-29; *Ørvig, 1973*, Text-fig. 1C; *Blais et al., 2011*, *Blais et al., 2015*; *Burrow et al., 2018*) also suggests the presence of a mixed dermal-oral signaling environment extending beyond the mouth. Together with these observations, the tooth-like ornament and the ornament-like teeth of *Lophosteus* call into question the demarcation between dermal and oral developmental domains.

The description of ornament invasion presented above incorporates the assumption that the position and orientation of the jawbone is static relative to the edge of the mouth. In fact, the bone appears to have rotated labially during growth (*Figure 2*), so that the location of the mouth margin shifted during ontogeny from the lingual founder ridge, via each lingual row of the first-generation teeth, to each successive row of replacement teeth at the current lingual margin of the oral lamina. This implies that the oral-dermal epithelial boundary does not drift lingually to any great degree; rather, the rotation of the bone causes the labial tooth rows to move onto the face, where they get covered by dermal epithelium and overgrown by ornament. This is comparable to the rotation of the tooth whorls at the jaw margin of primitive chondrichthyans, with the post-functional teeth slid beneath the skin (*Smith and Coates, 2001*, Figure 14.3(a); *Williams, 2001*).

## Developmental implications: alternate pattern throughout the oral and dermal domains

In *Lophosteus*, the first-generation teeth are added sequentially toward the growing lingual margin, with the tooth families arranged in horizontal alternate files. In the next stage, each successor, if not overgrown by invasive odontodes, turns into an initiator and sets up its own tooth family by cyclic replacement. The replacement teeth hence inherit the pattern of the first-generation teeth, generation after generation, forming vertical alternate columns. The replacement columns, including those inserted later and those disturbed by overgrowing odontodes, follow the same transverse alignment as the first-generation teeth (*Figures 4B*, *5* and *6*). Longitudinally, the exposed tooth rows appear somewhat irregular, which is due to two reasons. Firstly, new positions that are not established by the first-generation teeth are inserted whenever space is available along the marginal replacement column (e.g. *Figure 5B*, RC3-10). Secondly, different labial positions are terminated randomly by different generations of overgrowing odontodes, and the total number of replacement cycles is different in each tooth position. As a consequence, the final replacement teeth drifted lingually from their first-generation teeth for a variable distance, reflected by the variable length of replacement

columns (*Figure 6*, compare TR7-3 with TR 6–4 and TR4-4). Nevertheless, the marginal positions that maintain the original pattern throughout the growth of bone are invariably aligned in a row.

Sequential addition along the files established by the first generation also applies to the stellate odontodes on the facial lamina. Such an alternate arrangement is constant throughout the teeth and tooth-like odontodes on the marginal jawbone of *Andreolepis* and *Kosoraspis* as well, irrespective of whether they will be overgrown by younger generations of odontodes (*Chen et al., 2016*; *Vaškaninová et al., 2020*). The alternate pattern of teeth and ornament of *Lophosteus* is already established by the founder ridges. The ornament differs from teeth in the fact that every other alternate position of ornament is suppressed as the ornament extends labially (*Figure 4—figure supplement 1B*); this is because consecutive generations of dermal odontodes increase in length more quickly than the bone. The replacement of two teeth by a single larger successor at a crowded site has been observed in sharks, frogs and lizards, and considered as a common phenomenon (*Gillette, 1955*; *Edmund, 1960*; *Cooper, 1963*; *Osborn, 1971*; *Reif, 1976*). In alligator embryos, the reduction of jaw growth can cause the suppression of a tooth family during ontogeny (*Westergaard and Ferguson, 1987*). The suppression does not represent an irregularity; instead, it may reflect the fundamental mechanism producing the alternate pattern.

The hexagonal pattern is the most efficient form of close packing of rounded objects. The close packed teeth of myliobatid rays, which have turned into short hexagonal prisms (*Edmund, 1960*; *Underwood et al., 2015*), is an extreme example of a dentition imitating the structure of a honeycomb. A regular pattern, which was thought to be unique to teeth and reflect the spatio-temporal regulation of the dental lamina (*Smith, 2003*; *Underwood et al., 2016*), is actually not uncommon in dermal odontodes, as well as in bony denticles. Ordered tubercles can be seen covering the armor of jawed stem gnathostomes as long as there is only one generation, irrespective of whether they are dentinal units on the gnathal plates and spines, or bony units on the postbranchial lamina (*Bystrow, 1957*, Fig. 2; *Burrow, 2003*; *Johanson and Smith, 2003*; *Johanson and Smith, 1999*; *Young, 2003*). In the polyodontode scales of the earliest known chondrichthyans, odontodes are often organized in parallel or radial rows by sequential addition (*Andreev et al., 2020*).

An alternate pattern can be produced by odontodes that are laid down directly on the bony plate, simply through filling the gaps between the odontodes in the previous row, even if the previous row is from the older generation (*Figure 4—figure supplement 1B*). But it is difficult for the enlarged overgrowing odontodes to find and fit such gaps, by reason of the preexisting odontodes, so the pattern will be disturbed. This effect is clearly visible in a digitally dissected spine of the jawed stem gnathostome *Romundina*, which carries three generations of odontodes (*Jerve et al., 2017*, Fig. 2D1–D3); the first generation shows an ordered pattern, but this breaks down in the second and third generations. The single-file arrangement of the first-generation odontodes of *Andreolepis* scale is also obscured by the overgrowing odontodes (*Qu et al., 2013*).

All these examples agree with a fundamental embryologic mechanism of odontode patterning shared among the skeletons of vertebrates, which had evolved prior to the origin of teeth, as already proposed by *Osborn, 1971*. The alternate pattern can be self-generating, as long as the size of inhibitory zones is equivalent or in a smooth gradient (*Osborn, 1977*). Therefore, the regularity of organization should not be considered as a criterion of true teeth. The claim that the ectoderm (dermal epithelium) lacks patterning capacity (*Fraser and Smith, 2011*; *Smith and Johanson, 2015*), which has been used to support the idea of a fundamental difference between dermal odontodes and teeth, is biased by the derived adult condition of modern chondrichthyans. Actually, skin denticles of chondrichthyans embryos are added sequentially in regular rows, including the caudal denticles, dorsal denticles and the first-generation general denticles (*Grover, 1974*; *Reif, 1976*; *Ballard et al., 1993*; *Miyake et al., 1999*; *Johanson et al., 2007*; *Debiais-Thibaud et al., 2015*; *Maisey and Denton, 2016*; *Martin et al., 2016*; *Cooper et al., 2018*). The oro-pharyngeal denticles also emerge in rows in embryos, even if frequently interrupted by the oral papillae and undulated by the uneven surface of the oropharynx (*Rangel et al., 2017*, Fig. 4F,G). All the skin or oral denticles are likely to be self-organized by Turing's reaction-diffusion system, a patterning mechanism probably common to epithelial appendages (*Maisey and Denton, 2016*; *Cooper et al., 2018*). Later in ontogeny, these denticles may display a random organization, because the regular rows have been obscured by the mix of denticles in variable sizes, which may be due to the loss and repair of original denticles at different developmental stages. Even so, both skin and oropharyngeal denticles retain their alternating pattern in adulthood (*Rangel et al., 2016*). Denticle files with regular spacing line

up the rear border of the pharyngeal pads in some requiem sharks, remarkably resembling the tooth files lining the length of jaw and probably functioning like osteichthyan pharyngeal dentitions (*Nelson, 1970*, fig. 13, 15, 16B). Wound-healing experiments on sharks show that the loss of the diagonal rows and the rostro-caudal polarity as well as the regular size and shape of the scales in repaired squamations is actually caused by disturbance of the original diagonal arrangement of the anchoring collagen fibers (*Reif, 1978a*). By contrast, the orderly shedding and replacing of chondrichthyan teeth preserves the embryonic pattern.

## Evolutionary developmental relationship between teeth and ornament

Our data from *Lophosteus* are in two respects uniquely informative about the relationship between teeth and ornament. Firstly, as a Silurian stem osteichthyan, *Lophosteus* represents a very short phylogenetic branch in a basal part of the gnathostome crown group, and is thus likely to present primitive characters for the Osteichthyes and maybe for the Gnathostomata as a whole; secondly, we can trace the developing relationship between teeth and dermal odontodes through the life history of the animal, whereas all the dermal odontodes and teeth that have been compared in previous studies are fully differentiated forms in adults.

The unified arrangement of the teeth and ornament of *Lophosteus* challenges the currently popular idea that teeth and dermal denticles have fundamentally different patterning regimes (*Fraser and Smith, 2011*). In *Lophosteus*, teeth and ornament are never starkly different. The replacement teeth can have an ornament-like appearance, and can be partially shed and overlapped. The dermal odontodes can be added sequentially and alternately, and organized in rows and files. New tooth positions can be inserted once a gap appears, obliterating the original addition sequence.

Looking further afield, it is noteworthy that the 'extra-oral teeth' that occur in some teleosts also display a regular organization, development in epithelial invaginations, and shedding-replacing by basal resorption of the attachment bone and supporting bone (*Sire and Huysseune, 1996*; *Sire et al., 1998*; *Sire, 2001*; *Sire and Allizard, 2001*). These structures reveal the potential plasticity of odontodes and suggest the conventional criteria of 'true teeth' (*Burrow, 2003*; *Smith and Johanson, 2003a*; *Smith and Johanson, 2003b*) are in fact not unique to oral teeth.

Therefore, teeth and ornament are not merely homologs (*Reif, 1982*; *Huysseune and Sire, 1998*; *Donoghue, 2002*), as supported by a common gene regulatory network (*Fraser et al., 2010*; *Debiais-Thibaud et al., 2011*). More importantly, they develop from the same developmental module, modified through a simple mechanism of heterotopy (*Debiais-Thibaud et al., 2011*), which can be evidenced by the developmental continuum, as on the marginal jawbone of *Lophosteus*. However, contrary to the scale-to-teeth scenario of the 'outside-in' theory, we argue that teeth did not evolve from ontogenetically mature ornament, but rather from a primordial type of founder odontode, when covered by the oral epithelium. The primordial odontodes might be similar all over the body, outside or inside the oropharyngeal cavity, with later deposited odontodes differentiated according to their location and function. For example, the primordial odontodes in the ganoid scales of *Andreolepis* or the cosmoid scales of *Psarolepis*, revealed by PPC-SRμCT, have a more pointed and tooth-like morphology than the overgrowing odontodes (*Qu et al., 2013*; *Qu et al., 2017*), resembling the relationship between the founder ridges and ornament in *Lophosteus*. This earliest developmental stage of the odontode skeleton may have been lost in derived taxa or missed by the conventional investigative techniques. Direct comparison between the substantially modified mature subsets of odontodes, could lead to the conclusion that teeth and dermal odontodes are two wholly separate systems. Crucially, *Lophosteus* shows us that only ontogenetic data going back to the earliest stages of development are able to reveal the original patterning relationships within the odontode skeleton.

## Materials and methods

### Specimen collecting and photography

The marginal jawbone fragments of *Lophosteus* were collected from fallen blocks of limestone at Ohesaare cliff, Saaremaa Island, Estonia, the type locality of *Lophosteus*. Acetic acid dissolution and extraction of the microremains were carried out at the Department of Earth Sciences of Lund

University and the Department of Organismal Biology of Uppsala University, Sweden. The specimens are registered to the Geological Institute, Tallinn, Estonia GIT 760–12 ~ 28. All specimens in *Figure 3—figure supplement 1* were photographed under an Olympus SZX10 microscope-camera setup, with ImageView imaging software, at the Swedish Museum of Natural History, Stockholm.

## PPC-SRμCT

The specimen was imaged at beamline ID19 of the European Synchrotron Radiation Facility (ESRF) in Grenoble, France, using propagation phase-contrast synchrotron radiation microtomography (PPC-SRμCT) adapted to fossil mineralized tissues histology (*Tafforeau and Smith, 2008*; *Sanchez et al., 2012*). With an isotropic voxel size of 0.696 μm, the scan was obtained with an objective 10×, NA0.3 coupled with a 2 × eyepiece. The optics, associated with a gadolinium gallium garnet crystal of 10 μm thickness (GGG10) scintillator, is coupled to a FreLoN 2K14 detector [fast readout low noise camera; *Labiche et al., 2007*] used in full frame mode with a fast shutter. The specimen was set up 15 mm from the optics. The gap of the undulator U17.6 was set to 20 mm and provided a pink beam (direct beam with a single main narrow harmonic) at an energy of 19keV, filtered by 0.7 mm of aluminum. A total of 4998 projections of 0.3 s each were taken over 360° by half-acquisition of 600 pixels. Reconstruction was done with a modified version of a single-distance phase retrieval approach (*Paganin et al., 2002*; *Sanchez et al., 2012*).

The virtual thin sections of the sample in the form of stacks of images were segmented into three-dimensional sub-volumes through the software VG Studio 3.1. Embedded subtle structures, such as the surfaces of resorption and dentine, were traced manually.

## Acknowledgements

We thank ESRF for granting beamtime to the scanning proposal (ES151, DC) and the Knut and Alice Wallenberg Foundation for funding (Wallenberg Scholarship, PEA). We appreciate the helpful comments from Abigail Tucker, Qingming Qu, Jan Stundl, Rolf Ericsson, John Long, Zerina Johanson and two anonymous reviewers.

## Additional information

### Funding

| Funder | Grant reference number | Author |
| --- | --- | --- |
| Knut och Alice Wallenbergs Stiftelse | Wallenberg Scholarship | Per E Ahlberg |

The funders had no role in study design, data collection and interpretation, or the decision to submit the work for publication.

### Author contributions

Donglei Chen, Conceptualization, Software, Formal analysis, Investigation, Visualization, Writing - original draft, Writing - review and editing; Henning Blom, Conceptualization, Resources, Investigation, Visualization, Project administration, Writing - review and editing; Sophie Sanchez, Software, Methodology, Writing - review and editing; Paul Tafforeau, Data curation, Methodology, Writing - review and editing; Tiiu Märss, Resources, Writing - review and editing; Per E Ahlberg, Conceptualization, Resources, Supervision, Project administration, Funding acquisition, Visualization, Writing - original draft, Writing - review and editing

### Author ORCIDs

Donglei Chen (iD) https://orcid.org/0000-0002-0471-8162
Henning Blom (iD) https://orcid.org/0000-0002-1257-0057
Sophie Sanchez (iD) http://orcid.org/0000-0002-3611-6836
Paul Tafforeau (iD) http://orcid.org/0000-0002-5962-1683
Per E Ahlberg (iD) https://orcid.org/0000-0001-9054-2900

Decision letter and Author response
Decision letter https://doi.org/10.7554/eLife.60985.sa1
Author response https://doi.org/10.7554/eLife.60985.sa2

## Additional files

### Supplementary files
• Transparent reporting form

### Data availability

All synchrotron scan and segmentation data are available through the ESRF paleontology database. http://paleo.esrf.fr/picture.php?/5267/category/2933. Please note that free registration is required to access the data.

The following dataset was generated:

| Author(s) | Year | Dataset title | Dataset URL | Database and Identifier |
|---|---|---|---|---|
| Chen D, Blom H, Sanchez S, Tafforeau P, Märss T, Ahlberg PE | 2020 | *Lophosteus superbus*, marginal jawbone (GIT 760-12) | http://paleo.esrf.fr/picture.php?/5267/category/2933 | ESRF heritage database for palaeontology, evolutionary biology and archaeology, 5267/category/2933 |

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
