## [Decision Letter]

**Acceptance summary:**

This is a highly significant paper that sheds light on the origins of teeth and their relationship, as part of a single biological system, with dermal odontodes, and as such will be of much interest to the evolutionary biology community. Thank you for making the necessary changes to format as suggested and dealing with the reviewers comments efficiently.

**Decision letter after peer review:**

Thank you for submitting your article "The developmental relationship between teeth and dermal odontodes in the most primitive bony fish *Lophosteus*" for consideration by *eLife*. Your article has been reviewed by three peer reviewers, one of whom is a member of our Board of Reviewing Editors, and the evaluation has been overseen by George Perry as the Senior Editor. The following individual involved in review of your submission has agreed to reveal their identity: Zerina Johanson (Reviewer #2).

The reviewers have discussed the reviews with one another and the Reviewing Editor has drafted this decision to help you prepare a revised submission.

This paper gives a detailed and well-illustrated description of the histological evolution of odontodes in a stem Osteichthyan fish *Lophosteus* using high resolution synchrotron imagery. It shows that teeth and dermal ornamentation on dermal bones of the jaw evolved from the same odontode regulation system. As such it is a valuable contribution towards the current debate on the origins of teeth in vertebrates.

Revisions:

Materials and methods, Results and conclusions are clearly laid out. In essence most of the figures shows odontode generations and patterns of replacement and movement, but Figure 7 is the key summary diagram that encapsulates the real findings of the paper. Much of the text is heavily descriptive, telling how generations of odontodes develop and are replaced or migrate towards different zones. As such it provides strong and convincing evidence for the hypothesis. We feel that much of this would not be of interest to non-specialists and the general reader audience of *eLife*.

To shorten the paper please edit down the manuscript and cut out sections of the description text and some of the illustrations (eg Figures 4-6) and place all this in the Supplementary information, then edit the paper to make a shorter, more condensed version of the main findings (referring to S.I. for such data, extra figures etc). I think it would then be suitable for *eLife*, but not in its current long format. Please see the attached links to the annotated pdf revisions by reviewers 2, 3 for individual points to address in your revision.

Reviewer #2:

This is an important contribution from the Ahlberg lab, featuring the exquisitely detailed reconstructions of high-resolution synchrotron scans. The focus is on the stem-osteichthyan *Lophosteus*, following on from other research done by the team in recent years, on taxa such as *Andreolepis*. This manuscript substantially contributes to our growing body of knowledge of tooth and dermal denticle development in jawed vertebrates, and I was pleased to see the broader comparisons made in the manuscript, including stem-chondrichthyans.

One of my main questions statements regarding the suggested similarity between the teeth and dermal denticles in *Lophosteus*. However, the earliest stage shows two “founder ridges”, one related to the labial dermal denticles, and the other lingual, associated with the teeth. The first structures developing from these are described as being distinct, with this being lost soon after as the boundary between oral and dermal denticle regions in effect breaking down. At this point the teeth and denticles near the boundary take on each other's characteristics. I agree with this, but think that the earliest stage shows how distinct these are to begin with?

---

## [Author Response]

This paper gives a detailed and well-illustrated description of the histological evolution of odontodes in a stem Osteichthyan fish Lophosteus using high resolution synchrotron imagery. It shows that teeth and dermal ornamentation on dermal bones of the jaw evolved from the same odontode regulation system. As such it is a valuable contribution towards the current debate on the origins of teeth in vertebrates.Revisions:Materials and methods, Results and conclusions are clearly laid out. In essence most of the figures shows odontode generations and patterns of replacement and movement, but Figure 7 is the key summary diagram that encapsulates the real findings of the paper. Much of the text is heavily descriptive, telling how generations of odontodes develop and are replaced or migrate towards different zones. As such it provides strong and convincing evidence for the hypothesis. We feel that much of this would not be of interest to non-specialists and the general reader audience of eLife.To shorten the paper please edit down the manuscript and cut out sections of the description text and some of the illustrations (eg Figures 4-6) and place all this in the Supplementary information, then edit the paper to make a shorter, more condensed version of the main findings (referring to S.I. for such data, extra figures etc). I think it would then be suitable for eLife, but not in its current long format. Please see the attached links to the annotated pdf revisions by reviewers 2, 3 for individual points to address in your revision.

We have addressed these points as well as the criticisms of the reviewers. In addition, because we wanted to be sure that the paper is readable and informative to a developmental biology audience, we sent the manuscript out to two colleagues who work on dental development (Qingming Qu and Abigail Tucker) so that they could comment on it. Their views matched those of the two reviewers fairly closely, but we have made some additional changes that specifically reflect their wishes (notably, moving the explanation of the odontode numbering system from the Materials and methods to the main text, at the request of Abigail Tucker).

In order to shorten the manuscript, we have moved five figures plus figure legends to Supplementary Information. We have also moved some descriptive text into the figure legends of the main paper. We feel this works well, the figure legends are not overly large and it keeps the main text flowing well while simultaneously keeping the additional information more readily available for the reader than it would be in Supplementary Information.

Reviewer #2:This is an important contribution from the Ahlberg lab, featuring the exquisitely detailed reconstructions of high-resolution synchrotron scans. The focus is on the stem-osteichthyan Lophosteus, following on from other research done by the team in recent years, on taxa such as Andreolepis. This manuscript substantially contributes to our growing body of knowledge of tooth and dermal denticle development in jawed vertebrates, and I was pleased to see the broader comparisons made in the manuscript, including stem-chondrichthyans.One of my main questions statements regarding the suggested similarity between the teeth and dermal denticles in Lophosteus. However, the earliest stage shows two “founder ridges”, one related to the labial dermal denticles, and the other lingual, associated with the teeth. The first structures developing from these are described as being distinct, with this being lost soon after as the boundary between oral and dermal denticle regions in effect breaking down. At this point the teeth and denticles near the boundary take on each other's characteristics. I agree with this, but think that the earliest stage shows how distinct these are to begin with?

There are two separate stories here. The founder ridges are only slightly different from each other, but the first-generation ornament odontodes and teeth quickly become more distinct as they are added further and further away from these ridges. However, later on, when the ornament begins to invade the territory of the dentition, the teeth and ornament odontodes in the “invasion zone” come to resemble each other morphologically. This only happens where the two types of odontodes are in close proximity, and we interpret it as the result of a mixed molecular signaling environment.